



# Surface and top-of-atmosphere radiative feedback kernels for CESM-CAM5

Angeline G. Pendergrass[1], Andrew Conley[1], Francis Vitt[1]

[1]National Center for Atmospheric Research, Boulder, CO, 80305, USA

*Correspondence to*: Angeline G. Pendergrass (apgrass@ucar.edu)

**Abstract.** Radiative kernels at the top of atmosphere are useful for decomposing changes in atmospheric radiative fluxes due to feedbacks from atmosphere and surface temperature, water vapor, and surface albedo. Here we describe and validate radiative kernels calculated with the large-ensemble version of CAM5, CESM 1.1.2, at the top-of-atmosphere and the surface. Estimates of the radiative forcing from greenhouse gases and aerosols in RCP8.5 in the CESM large ensemble

simulations are also included. As an application, feedbacks are calculated for the CESM large ensemble. The kernels are freely available at https://zenodo.org/record/997902, and accompanying software can be downloaded from https://github.com/apendergrass/cam5-kernels.

## 1 Introduction

A *radiative feedback kernel* is the radiative response to a small perturbation in, e.g., temperature or water vapor. Radiative

feedback kernels for the top-of-atmosphere (TOA) are useful for decomposing changes in atmospheric radiative fluxes due to feedbacks from atmosphere and surface temperature, water vapor, and surface albedo (Soden and Held, 2006). Radiative kernels at the surface or in the atmospheric column are useful for decomposing the changes in precipitation (Pendergrass and Hartmann, 2014; Previdi, 2010).

Widely used TOA radiative kernels were calculated with the GFDL model (Soden and Held, 2006). Other kernels include those from CAM3 (Shell et al., 2008) and more recently from the MPI-ESM-LR model (Block and Mauritsen, 2013). The only publicly available surface radiative kernels are from ECHAM5 (Previdi, 2010) and MPI-ESM-LR, which is a more recent version of the ECHAM model (discussed in Fläschner et al., 2016). Not all kernels have been validated to test the accuracy to which total radiative fluxes from a model simulation can be recovered with kernel calculations; examples that

have been validated against model calculations of radiative flux due to doubling of carbon dioxide include Shell et al (2008) and Block and Mauritsen (2013).

Here we describe and validate radiative kernels calculated with CESM-CAM5 (Hurrell et al., 2013) for the top-of-atmosphere and the surface. These radiative feedback kernels were calculated with CESM version 1.1.2, the same which was



used for the 40-member CESM large ensemble (Kay et al., 2015). The TOA kernels are an update from CAM3 (Shell et al., 2008). We also include estimates of radiative forcing due to greenhouse gases and aerosols in RCP8.5 in the CESM large ensemble simulations, which are necessary for calculating the cloud feedback using radiative kernels.

## 2 Calculations

In order to calculate the radiative feedback kernels, we make offline radiative transfer calculations following the methodology of Soden and Held (2006) with the Parallel Offline Radiative Transfer code (PORT; Conley et al., 2013), updated for compatibility with CAM5 microphysics. We re-integrate the first member of the CESM large ensemble for one year to obtain temperature, mixing ratio, and other necessary fields to run offline radiative calculations, writing out instantaneous fields every 3 hours. All calculations were completed on NCAR's Yellowstone computer system
(Computational and Information Systems Laboratory, 2012).

### 2.1 Radiative kernels

Together, all calculations consumed approximately 200,000 core hours on NCAR's Yellowstone supercomputer. The limiting factor for throughput is the size of PORT input data: three-hourly 3-D temperature, moisture, and cloud fields; about
7.5 TB of disk space are needed to run one month of three-hourly PORT calculations for each vertical level of each kernel, with 63 global radiative transfer computations per kernel-month. The TOA radiative kernels are shown in Fig. 1, and surface kernels in Fig. 2. The atmospheric column kernel is the difference between these two kernels. The procedure to produce each kernel follows.

*Atmospheric temperature kernel*
To calculate the atmosphere temperature kernel, we perturb the air temperature by 1 K in each hybrid sigma-pressure level at a time, for each 3-hourly instantaneous field, and take the difference of the TOA and surface radiative fluxes from the control in response to each perturbation. PORT is adjusted so that the hygroscopic growth of aerosols is not affected by the modified temperatures and water vapor concentrations. The calculations are carried out in CESM's hybrid sigma-pressure
vertical coordinate and has units of W m$^{-2}$ K$^{-1}$ level$^{-1}$. These hybrid sigma-pressure radiative feedback kernels can be interpolated onto standard CMIP pressure levels (see the description of example code in Section 5), as is done for display purposes here; the atmospheric temperature kernel is shown in Fig. 1a,c for the TOA and Fig. 2a,c for the surface.

*Surface temperature kernel*



In CESM, the upwelling longwave flux at the surface is indirectly related to the surface temperature. To calculate the surface contribution to the temperature kernel, we perturb upwelling surface longwave radiation by an amount consistent with 1 K warming at constant effective emissivity. The surface temperature kernels are shown in Figs. 1,2e.

*Atmospheric moisture kernel*

The atmospheric moisture kernel is constructed by perturbing the mixing ratio on each hybrid sigma-pressure level by the amount that would result in constant relative-humidity moistening if there were a warming of 1 K. Saturation mixing ratio is calculated for each 3-h instantaneous state using the mixhum_ptd function in NCL (UCAR/NCAR/CISL/TDD, 2015), which calculates saturation with respect to liquid water following List (1951). Code to closely approximate the perturbation with

10 monthly-mean fields using this NCL calculation is provided. As with the temperature kernel, the mixing ratio does not change for the purpose of aerosol radiative properties (hygroscopic growth is held invariant). The moisture kernel is shown in Figs. 1,2 b,d,f,h.

*Surface albedo kernel*

The surface albedo kernel is the change in radiative flux for a 1 % change in surface albedo. The calculation is carried out by perturbing the direct and diffuse shortwave albedos, asdir and asdif, simultaneously by 1 % each. The surface albedo kernel is shown in Figs. 1,2g.

## 2.2 Forcing

To estimate the radiative forcing, we re-integrate CESM for the year 2096, writing out the fields needed for PORT calculations every 3 hours. The baseline for the forcing calculations is the same 2006 control as the kernel calculations. The greenhouse gas and aerosol forcing are shown in Figure 3.

*Greenhouse gas forcing*

We use the greenhouse gas concentrations (carbon dioxide, methane, CFCs, $N_2O$, and ozone) from 2096 with the tropospheric temperature, mixing ratio, and other radiatively-relevant fields from 2006. To account for stratospheric adjustment, we use the 2096 stratospheric temperature and water vapor mixing ratio in the calculation. The tropopause is defined as 100 hPa at the equator and 300 hPa at the poles and varies by cosine of latitude in between, following (Soden and Held, 2006). The resulting estimate approximates the radiative forcing following the definition of Myhre et al., (2013),

though it includes adjustment of water vapor as well as temperature in the stratosphere.

*Aerosol forcing*



To calculate the aerosol forcing, we apply black carbon, sulfate, secondary organic aerosol, primary organic matter, dust, sea salt, and aerosol temperature and mixing ratio from 2096, with temperature, mixing ratio, greenhouse gas and all other fields from 2006, with no adjustments to the stratosphere. The resulting estimate is the instantaneous radiative forcing.

### 3 Validation

Radiative kernels can create a useful but approximate decomposition the contributions to changes in radiative fluxes. Application of radiative kernels assumes that changes in radiative fluxes are linear in changes in constituents, and that the response to changes in temperature and moisture at different vertical levels are independent.

In order to quantify these sources of error associated with the kernel calculations, we calculate the radiative response to the
changing surface albedo and temperature and tropospheric moisture and temperature from 2006 to 2096, holding all other fields fixed (including clouds and forcing agents). The change in global, annual mean clear-sky radiative fluxes is shown in Fig. 4 for the TOA and in Fig. 5 for the surface.

At the TOA, errors in LW and SW range from 14-26% of the total change in radiative flux. Because the LW and SW offset
each other, net absolute flux changes are smaller than either LW or SW individually, and so the net relative errors are larger. At the surface, LW errors range from 15-24%. The changes in absolute SW surface downwelling flux are small (because the clouds are fixed in the validating simulations), and so the relative error in all-sky surface flux is large.

### 4 Known issues

Because the kernel calculation is computationally intensive, it is based on just one year. We can estimate the effect of interannual variability – our choice of a single year for the kernel calculation – on the error by examining the spread in total radiative flux change across the 40 member large ensemble (Table 1). We make this estimate indirectly, based on changes in TOA radiation rather than feedback components to avoid additional computation. The standard deviation across the ensemble is $0.4 - 0.5$ $Wm^{-2}$ for LW fluxes and $0.2 - 0.4$ $Wm^{-2}$ for SW fluxes. The error due to interannual variability is
smaller than the error due to nonlinearity at the TOA for both LW and SW and clear-sky and all-sky conditions (c.f. Fig. 4), and in the LW at the surface (c.f. Fig. 5). For the surface SW response, the error due to interannual variability and nonlinearity are similar in magnitude (though the absolute response is small).

In standard applications that make use of experiments from models other than CESM(CAM5), errors will also arise due to
differences in radiative transfer codes between models, (e.g., DeAngelis et al., 2015; Soden et al., 2008); we do not quantify



these errors explicitly here, but we do compare feedback decompositions made with other radiative kernels from the literature in Section 5.

**5 Application**

Next we show a sample application of the kernels to the CESM large-ensemble simulations. In order to apply the kernels, one needs monthly, long-term mean water vapor mixing ratios, as well as changes in temperature, mixing ratio, and surface albedo. To calculate the cloud feedback, change in cloud radiative effect is also required.

We apply the radiative kernels to the CESM large ensemble integrations to diagnose the changes in top-of-atmosphere
radiative feedbacks and surface radiative flux changes (Figure 6 and Table 2). The changes in surface and tropospheric temperature, water vapor mixing ratio, surface albedo, and cloud radiative effect are calculated from 30-year averages for each month from each ensemble member, 1976-2005 and 2071-2100. The cloud feedback calculation follows Soden et al., (2008).

Table 2 shows a comparison between feedback values calculated here and those diagnosed with other kernels and model simulations. Soden et al., (2008) report radaitive feedbacks from 14 CMIP3 m(Pendergrass, 2017a)odel simulations using three different sets of radiative kernels from CAM3, GFDL, and CAWCR kernels. The Planck response agrees closely. Water vapor and albedo feedbacks are both within 0.2 $Wm^{-2}K^{-1}$. The cloud feedback differs by only 0.1 $Wm^{-2}K^{-1}$, despite the fact that Soden et al., (2008) do not account account for aerosol radiative forcing. The only notable disagreement is in the
lapse rate feedback by 0.4 $Wm^{-2}K^{-1}$. Because the Planck feedback agrees closely between the two calculations, the difference is probably not due to the temperature kernel. Instead, it is probably caused by differing upper tropospheric temperature amplification between the CMIP3 and CESM large ensemble simulations. Block and Mauritsen (2013) report radiative feedbacks using MPI-ESM-LR kernels applied to abrupt carbon dioxide quadrupling experiments with the same model (they compare kernels calculated from different base states and also short and long timescales; we compare with their control base
state kernels and feedbacks evaluated on long timescales because this is most similar to our calculations). There is remarkably close agreement for all feedbacks, excepting only the water vapor feedback. This could arise from the kernels or from the change in water vapor in the simulations.

The variability of global-mean feedbacks across the CESM single-model ensemble provides an estimate of the magntiude of
30 internal variability on feedback magnitude. It is quite small. The largest standard deviation of any feedback is 0.02 W $m^{-2}$ K$^{-1}$, in contrast to multi-model ensembles (e.g., Soden et al., 2008), indicating that nearly all of the variability in feedbacks



across multi-model ensemble is due to structural uncertainty (the differences in formulation of different climate models) rather than internal variability.

## 6 Data and code

The provided dataset includes the four radiative kernels and the two radiative forcing files. Data are provided on the CESM
hybrid-sigma grid for comparison with CESM calculations. The datasets include net all-sky and clear-sky radiative fluxes, at both the top-of-atmosphere and surface. The sign convention is the same as CESM's: shortwave fluxes are positive downward, and longwave fluxes are positive upward. Sample temperature, moisture, surface radiative fluxes, and surface pressure are also included, as well as sample code to facilitate use of the kernels.

The data and code to calculate TOA temperature, water vapor, and albedo feedbacks are available for immediate download at https://zenodo.org/record/997902 and through ESGF at https://www.earthsystemgrid.org/dataset/ucar.cgd.ccsm4.cam5-kernels.html (Pendergrass, 2017b). The files included are listed in Table 3. Additional software tools – to regrid the kernels to pressure levels (including CMIP standard levels), calculate TOA Planck, lapse rate, and cloud feedbacks – are available at https://github.com/apendergrass/cam5-kernels.

*Acknowledgements*

William Frey and Ryan Kramer provided valuable feedback validating the kernels, testing and debugging code. A.G.P. was supported by an NCAR Advanced Studies Postdoctoral Research Fellowship and by the Regional and Global Climate Modeling Program (RGCM) of the U.S. Department of Energy's, Office of Science (BER), Cooperative Agreement DE-FC02-97ER62402. NCAR is sponsored by the National Science Foundation. Computing resources were provided by the
Climate Simulation Laboratory at NCAR's Computational and Information Systems Laboratory (CISL), sponsored by the National Science Foundation and other agencies.

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



**Table 1: Interannual variability in radiative fluxes.** The standard deviation of global, annual mean radiative flux changes across the 40 members of the CESM large ensemble.

|  | TOA | Surface |
|---|---|---|
| LW Total-sky | 0.46 | 0.38 |
| LW Clear-sky | 0.38 | 0.40 |
| SW Total-sky | 0.31 | 0.39 |
| SW Clear-sky | 0.17 | 0.21 |

**Table 2: Comparison of TOA radiative feedbacks.** TOA radiative feedbacks ($Wm^{-2}K^{-1}$) averaged over 40 CESM large ensemble simulations diagnosed with CAM5 radiative kernels here, compared against those from CMIP3 model simulations diagnosed with three different kernels as reported by Soden et al., (2008), and MPI-ESM-LR control state kernels and years 21-150 of abrupt carbon dioxide quadrupling simulations from the same model (Block and Mauritsen, 2013).

| Feedback | Here | Soden et al., (2008) | Block and Mauritsen (2013) |
|---|---|---|---|
| Planck | -3.2 | -3.1 or -3.2 | -3.19 |
| Lapse rate | -0.58 | -1 | -0.64 |
| Water vapor | 2.1 | 1.9 | 1.79 |
| Albedo | 0.51 | 0.3 | 0.48 |
| Cloud | 0.66 | 0.77 | 0.62 |

**Table 3: Included data files.** Data files comprising the dataset.

| Filename | Size | Units | Description |
|---|---|---|---|
| alb.kernel.nc | 20 MB | $W\ m^{-2}\ \%^{-1}$ | Albedo kernel |
| ts.kernel.nc | 20 MB | $W\ m^{-2}\ K^{-1}$ | Surface temperature kernel |
| t.kernel.nc | 608 MB | $W\ m^{-2}\ K^{-1}\ level^{-1}$ | Air temperature kernel |
| q.kernel.nc | 1.2 GB | $W\ m^{-2}\ K^{-1}\ level^{-1}$ | Moisture kernel |
| ghg.forcing.nc | 41 MB | $W\ m^{-2}$ | Greenhouse gas forcing |
| aerosol.forcing.nc | 41 MB | $W\ m^{-2}$ | Aerosol forcing |
| PS.nc | 5.1 MB | Pa | Surface pressure |





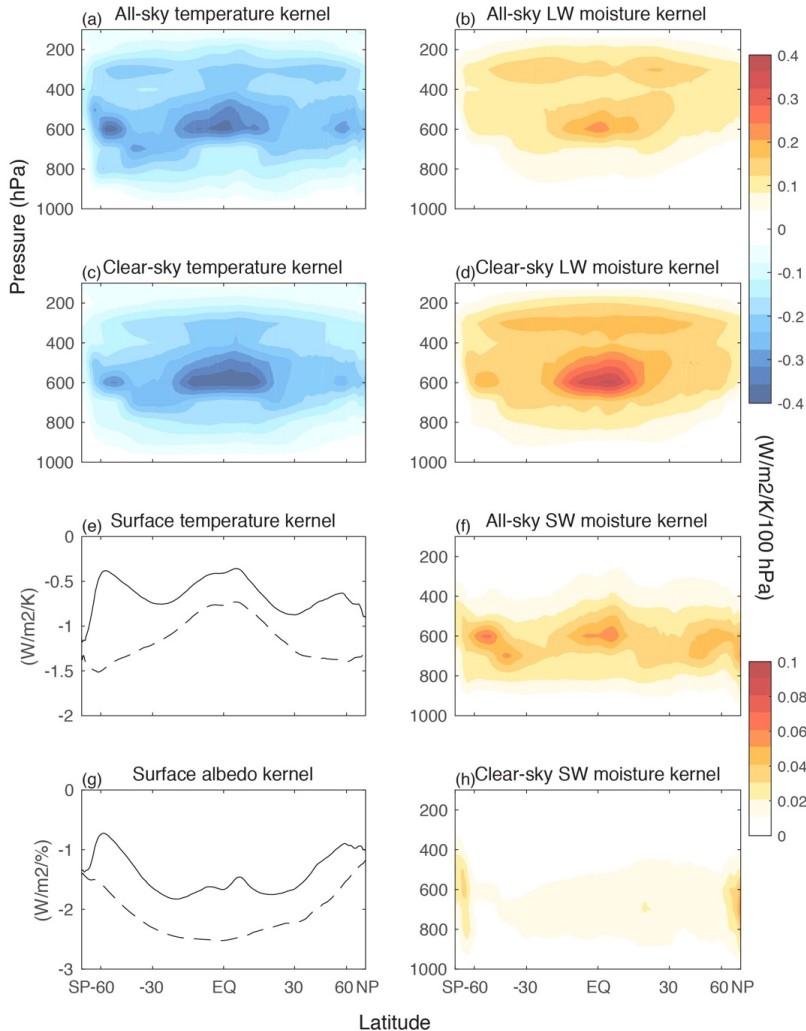

**Figure 1: Top-of-atmosphere kernels from CESM1(CAM5).** In panels (e) and (g) all-sky kernels are shown in solid and clear-sky dashed. The sign convention is positive downward. Solid lines are all-sky and dashed lines are clear-sky in (e) and (g).





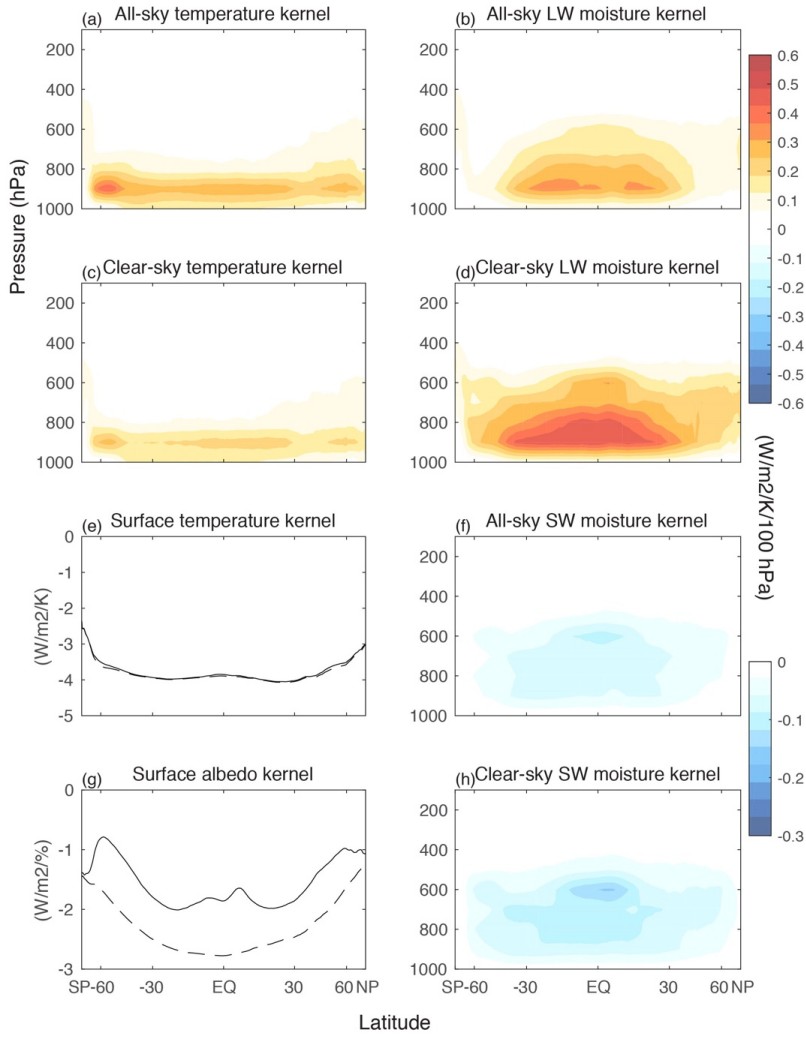

**Figure 2: Surface kernels from CESM(CAM5).** Zonal, annual mean temperature, longwave moisture and shortwave moisture kernels for all-sky and clear-sky. In panels (e) and (g) all-sky kernels are shown in solid and clear-sky dashed. The sign convention is positive downward.




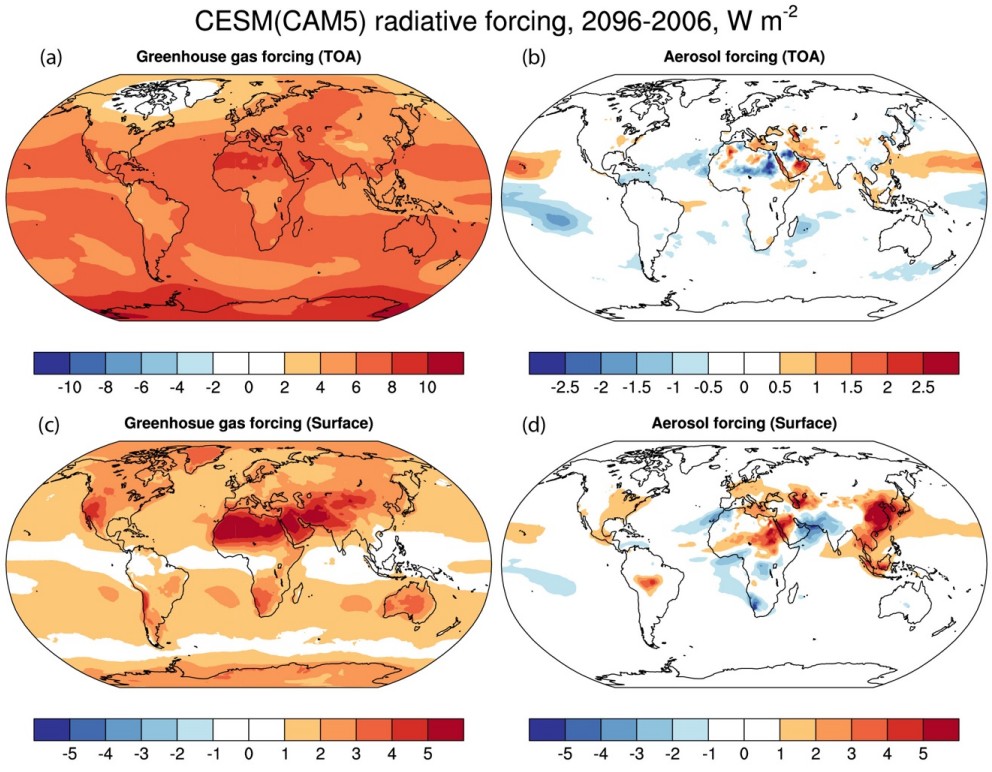

**Figure 3: Radiative forcing.** Net radiative forcing under the RCP8.5 scenario diagnosed from CESM(CAM5) for greenhouse gases (left) and the direct aerosol radiative forcing (right) at the TOA (top) and surface (bottom).




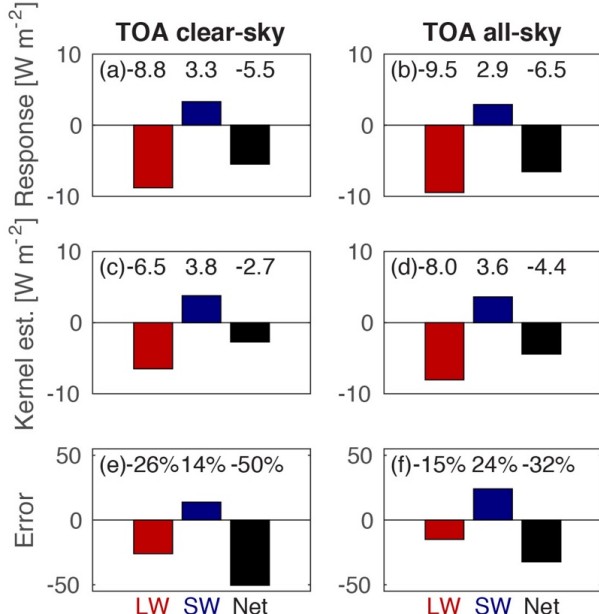

**Figure 4: Validation of TOA kernels.** TOA clear-sky and all-sky radiative field comparison (excluding cloud contributions). Errors are relative between the kernel estimate and total response.





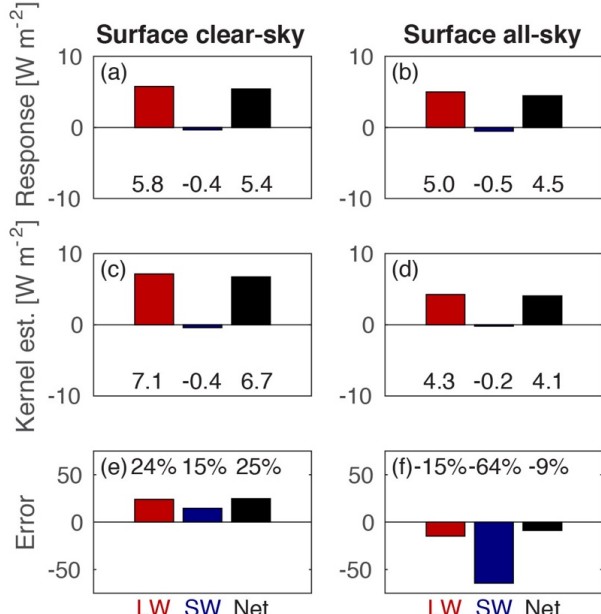

**Figure 5: Validation of surface kernels.** Surface clear-sky and all-sky radiative field validation (excluding cloud contributions).

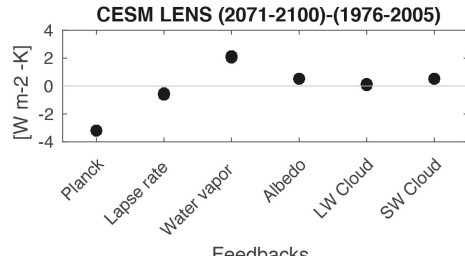

**Figure 6: CESM Large Ensemble kernels.** Feedback calculation for the CESM 40-member large ensemble using the TOA kernels.