# Peer review of "Surface and top-of-atmosphere radiative feedback kernels for CESM-CAM5"

_Earth System Science Data, 2017_

## Referee Comment (RC1) · Anonymous Referee #1 · 11 Nov 2017

The authors present a concise manuscript documenting and describing radiative feedback kernels for the top-of-atmosphere and surface produced with the CESM-CAM5. These kernels are likely to contribute to scientific advancement since they are one of few sets of documented kernels that also provide surface radiative responses and because they are produced from a different model and radiation code (although some clarification may be needed, as seen in detailed comments later). The authors also provide an assortment of example scripts that can be used for application of the kernels, which will be very helpful to the community. Nonetheless, I think the paper/data can be improved with regard to a few general things:

1) Some of the details in producing the kernels and applying them are lost due to the very concise nature of the text. While some of these details may be obvious to those

with experience in producing/using kernels, there is not quite enough basic information and detail for first-time kernel users. The authors provide several references pointing toward more information, but I don't think it would hurt or greatly increase the length of the text to give some of that information here. See specific comments below.

2) The provided data and scripts are a bit disorganized. If possible, I recommend putting all of the relevant data and example code in one place, rather than spanning two webpages. It may also help to organize the files into folders or categories based on their type (e.g., kernel files, forcing, relevant code, demos). On a more technical note, the readme file currently on the zenodo webpage should point to the actual filenames of all scripts that are described under "Additional scripts. . ."

Below are a series of specific comments, related to both scientific content and presentation.

Section 2, page 2, L6: It appears this radiation scheme is based on the CAM4, which is quite different and somewhat outdated compared to that in CAM5. Was the choice of this scheme because no other currently available kernels make use of it (e.g., I believe kernels based on ECHAM6 use a similar radiation scheme as CAM5), thereby increasing structural diversity among available kernels? It would be helpful to say a bit more here with regard to this.

Section 2, page 2, L8: Please explicitly state here what these "other necessary fields" are.

Section 2.1, first paragraph: State here that the simulations are only conducted for one year.

Section 2.1, L16: It is unclear where the number 63 comes from.

Section 2.1, L21-22: It should be mentioned that these perturbations are done at each grid cell.

Section 2.1, L22: Please clarify if these perturbations are computed with respect to

each corresponding control 3-hour timestep.

General computation comment: When going from the full kernel field (3-hourly, grid cell, level resolved) to the global averages presented here in tables, what is the procedure? For example, flux responses are first averaged in time (to monthly then annual) and space, and then vertically integrated with pressure weighting? I feel this should be mentioned.

Fig. 1 caption: This caption has some repetition and missing information. The Fig. 2 caption is fine and should be replicated for Fig. 1 but with obvious modifications.

Section 2, general: Can you expand a bit on how the patterns shown in the figures come about physically (e.g., the larger temperature kernel magnitudes in the tropics and the multi-peak structure in Fig. 1e), or at least provide references to previous work that has done so?

Fig. 3 caption: Should note that all panels are showing LW+SW, assuming that is actually the case.

Section 3, paragraph 2: It's a bit fuzzy what is done here. Are the kernels multiplied by the responses of the changing fields between the two years (with a procedure analogous to section 5)? Please expand. Also, explicitly state that these calculations are compared with the model-calculated flux responses between the same years.

Figures 4 and 5 captions: State that what is being shown is the global mean.

Section 3, paragraph 3 and related figures (4 and 5): Wouldn't the model-produced all-sky flux responses between 2006 and 2096 include cloud feedbacks as well? If so, what is the value of comparing those responses with corresponding all-sky kernel estimates that do not include the cloud changes (as stated on L11)?

Section 4, L22-23: change "based on changes in TOA..." to "based on model-computed changes in TOA..."
Section 4, L24: Insert "approximately" or symbols, e.g., "∼", in front of these values.

Section 5, first paragraph: It would be useful to 1) give some brief qualitative discussion about how these calculations are conducted (e.g., that the kernels are first monthly averaged then multiplied by monthly-resolved climatological changes in the fields) and 2) Include here a reference to section 6 where one can find more information about relevant example code.

Section 5, L6: Could you clarify why one needs long-term mean water vapor mixing ratios?

Table 2 caption: Suggest changing the "Here" heading to "CAM5"

Section 5, L25: Unsure why long timescales are more similar to your calculations since your kernels are only computed for one year. Please explain.

Section 5, L30, with regard to "0.02," perhaps also specify which feedback this is and give this value as a percentage of the mean as well. I also recommend reiterating here that this estimate does not account for potential variability among kernels themselves if computed from different years/ensemble members.

Section 6, L4: Please again specify what the four radiative kernels are. Also note that they are provided as monthly averages in the files.

Typos/grammar:

Section 2, page 2, L25: change "has units" to "have units"

Section 3, page 4, L5: insert "of" between "decomposition" and "the"

Section 3, page 4, L6: Suggest changing "linear in..." to "linear with respect to..."

Section 5, page 5, L9-10: Should "the changes in top-of-atmosphere radiative feed-backs" just be "the top-of-atmosphere radiative feedbacks"?

Section 5, page 5, L16: "m(Pendergrass, 2017a)odel"

---

## Referee Comment (RC2) · Anonymous Referee #2 · 16 Nov 2017

This paper presents a new set of radiation kernels that can be used for feedback analysis. This affords a useful means for GCM inter-comparison and for understanding their differences in climate sensitivity. I recommend publication after the following issues are properly addressed.

Two major suggestions are: 1. Provide comparisons between this new set of kernels and other kernels and note their differences. This would serve the community greatly to understand whether (and how) feedback determined using the kernel method is sensitive to the kernel dataset used.

2. Document more extensive validation tests. Currently only global mean values from one case (2096-2006) are reported (Fig. 4, 5); it is not clear whether the percentage errors reported are representative. Error statistics based on global maps of radiation

changes and time series of global means would make a more rigorous assessment.

Additional comments Page 1, Line 21. It is not a correct statement. New TOA and surface kernels computed from EARi atmosphere have been made available by Huang et al. (2017). This should be referenced here.

Page 2, Line 7. Note there is strong inter-annual variations of atmospheric states, e.g., El Nino vs. La Nina, which affects kernel values quantitatively. In relation to Suggestion 2 above, it is worth discussing, and if possible demonstrating, (in)accuracy in feedback, e.g., in the central tropical Pacific region, related to this issue.

Fig. 1. I am surprised to see the temperature sensitivity maximize in the middle tropical troposphere ($\sim$600hPa) here, which is noticeably different from other kernel datasets (e.g., Soden et al. 2008, Huang et al. 2017) - that may be due to misplaced clouds in CAM?

Fig. 2. I am also surprised by that the surface radiation sensitivity to temperature and humidity doesn't maximize in the lowermost atmospheric layer. These results do not agree with radiative transfer-based expectations.

References Huang, Y., Y. Xia and X. Tan, (2017), On the pattern of $CO_2$ radiative forcing and poleward energy transport, J. Geophys. Res.-Atmos., doi: 10.1002/2017JD027221

---

## Author Comment (AC1) · 9 Jan 2018

We thank both Reviewers for their attention to our manuscript and for their feedback. We have revised the manuscript, and we think it is substantially improved, thanks in part to the Reviewers' comments. Please find our responses below, as well as a version of the manuscript with tracked changes as a supplementary attachment.

**1  Response to Anonymous Reviewer 1**

We thank Reviewer 1 for their attention to our manuscript and for their feedback. We have incorporated your feedback and expanded the validation, and hopefully improved

the manuscript. In this document, your comments are italicized and our responses are in plain type.

*The authors present a concise manuscript documenting and describing radiative feedback kernels for the top-of-atmosphere and surface produced with the CESM-CAM5. These kernels are likely to contribute to scientific advancement since they are one of few sets of documented kernels that also provide surface radiative responses and because they are produced from a different model and radiation code (although some clarification may be needed, as seen in detailed comments later). The authors also provide an assortment of example scripts that can be used for application of the kernels, which will be very helpful to the community. Nonetheless, I think the paper/data can be improved with regard to a few general things:*

*1) Some of the details in producing the kernels and applying them are lost due to the very concise nature of the text. While some of these details may be obvious to those with experience in producing/using kernels, there is not quite enough basic information and detail for first-time kernel users. The authors provide several references pointing toward more information, but I don't think it would hurt or greatly increase the length of the text to give some of that information here. See specific comments below.*

Thank you for your constructive feedback about how to improve the text, in particular places where more information for those unfamiliar would be useful.

*2) The provided data and scripts are a bit disorganized. If possible, I recommend putting all of the relevant data and example code in one place, rather than spanning two webpages. It may also help to organize the files into folders or categories based on their type (e.g., kernel files, forcing, relevant code, demos). On a more technical note, the readme file currently on the zenodo webpage should point to the actual filenames of all scripts that are described under "Additional scripts. . ."*

We would of course also prefer a consolidated location for the data and the software,

but unfortunately we do not have access to a single repository that will provide long-term persistence for both data and software that meets the needs of our project. Specifically, we feel it is important to be able to include updates to the software as it is disseminated and the users find bugs. Github is an appropriate place to do this. However, the Github repository does not allow uploads of greater than 2 GB per project, and our data are 2.6 GB. The repository where the data are provided, the Earth System Grid (ESG), is widely used in the climate modeling community, but it is not possible to provide updates to the software there, so it is not an appropriate place to keep the software – though we do include a minimal set of scripts there, along with the data.

The README file included with the Zenodo data link does not contain the exact names of scripts that are found in the Github repository because the software repository can be updated over time, while the data respository cannot. For example, in response to offline comments associated we have included an additional script. We will continue to update the scripts in the event that users find additional code bugs.

*Below are a series of specific comments, related to both scientific content and presentation.*

*Section 2, page 2, L6: It appears this radiation scheme is based on the CAM4, which is quite different and somewhat outdated compared to that in CAM5. Was the choice of this scheme because no other currently available kernels make use of it (e.g., I believe kernels based on ECHAM6 use a similar radiation scheme as CAM5), thereby increasing structural diversity among available kernels? It would be helpful to say a bit more here with regard to this.*

The radiation scheme used for these kernels is RRTMG. It is used in CAM5, but not CAM4. Thank you for pointing out this ambiguity in our text, which probably arose because the published article about PORT that we cite was based on CAM4. PORT was originally developed for CAM4, and here we use a version that was ported to CAM5. The original manuscript stated that the CAM5 microphysics were ported (this was a

substantial technical challenge) and though we did not state it in the first manuscript version, the radiation scheme was also ported (but this was straightforward and not as memorable). We now state this explicitly and also cite the RRTMG radiation scheme.

*Section 2, page 2, L8: Please explicitly state here what these "other necessary fields" are.*

To calculate the kernels, the fields that are varied are temperature, mixing ratio, and four surface albedo parameters. For the radiative forcing calculations, carbon dioxide, methane, and a set of aerosol fields are also varied. Running the radiative calculation also requires the full instantaneous model 3-d atmospheric state to run, including clouds. This is now stated explicitly in the text.

*Section 2.1, first paragraph: State here that the simulations are only conducted for one year.*

We now include this statement.

*Section 2.1, L16: It is unclear where the number 63 comes from.*

The control, surface temperature, and albedo are each one kernel-month of calculation, and air temperature and moisture are each 30 (one per level). This is stated in the revised text.

*Section 2.1, L21-22: It should be mentioned that these perturbations are done at each grid cell.*

We now include this.

*Section 2.1, L22: Please clarify if these perturbations are computed with respect to each corresponding control 3-hour timestep.*

They are computed as monthly-mean differences. However, it should be noted that because these are absolute differences and averaging and differencing are linear operations (addition and subtraction, with division only by a constant), the order in which the averaging is done has no effect on the result. The order is chosen for computational efficiency (holding active storage as few 3-hourly instantaneous output fields as possible). The monthly mean difference of three-hourly perturbed and control fluxes is exactly equal to the difference of monthly mean three-hourly perturbed and control fluxes. Nonetheless, this is now included in the text.

*General computation comment: When going from the full kernel field (3-hourly, grid cell, level resolved) to the global averages presented here in tables, what is the procedure? For example, flux responses are first averaged in time (to monthly then annual) and space, and then vertically integrated with pressure weighting? I feel this should be mentioned.*

As discussed above, averaging absolute quantities in space and time is a linear operation, and this is also relevant here - the order of space and time averaging does not affect the result. Operations relevant to application of the kernels that are not interchangeable are, indeed, the pressure weighting (when pressure-coordinate kernels are used), and also normalization by global-mean surface temperature change. Pressure weighting is implicit to the kernels on the native CESM hybrid-sigma coordinate and is only employed when the kernels are interpolated to pressure levels. Pressure weighting is relevant for Figures 1 and 2, where the zonal-mean vertical structure of the kernels is presented, and no other figures or tables. Soden et al (2008) and Block and Mauritsen (2013) may have used pressure weighting in their calculations (which we reproduce in Table 2). As for normalizing by global mean surface temperature change, this is done in the final feedback calculations presented in Table 2 and Figure 6.

In the revised manuscript, we now state, "The global, annual mean change in radiative flux for each feedback is calculated and then normalized by the change in global-mean

surface temperature for each ensemble member; then, the ensemble average is calculated."

To describe the procedure for calculating Figures 1 and 2, the revised manuscript states, "To create these plots, the kernels are regridded to standard CMIP5 pressure levels in the troposphere including, pressure weighting (a vertical regridding script is available at https://github.com/apendergrass/cam5-kernels), and then the zonal and annual means are calculated."

*Fig. 1 caption: This caption has some repetition and missing information. The Fig. 2 caption is fine and should be replicated for Fig. 1 but with obvious modifications.*

This seems to have been a proofreading error. Thank you for pointing it out; we have now added the additional information to the Figure 1 caption.

*Section 2, general: Can you expand a bit on how the patterns shown in the figures come about physically (e.g., the larger temperature kernel magnitudes in the tropics and the multi-peak structure in Fig. 1e), or at least provide references to previous work that has done so?*

We avoided interpreting our data because this is beyond the scope of ESSD. We now provide two references to previous work that discusses the patterns.

*Fig. 3 caption: Should note that all panels are showing LW+SW, assuming that is actually the case.*

"Net" radiative forcing is LW+SW. We have now included LW+SW parenthetically to be more clear.

*Section 3, paragraph 2: It's a bit fuzzy what is done here. Are the kernels multiplied by the responses of the changing fields between the two years (with a procedure anal-*

[Figure]

*ogous to section 5)? Please expand. Also, explicitly state that these calculations are compared with the model-calculated flux responses between the same years.*

We have overhauled the validation in the revised manuscript, both in response to your comments as well as those from Reviewer 2. We have revised this paragraph so that the validation procedure is more clear, as suggested.

*Figures 4 and 5 captions: State that what is being shown is the global mean.*

These figures no longer appear in the manuscript; analogous information is now available in the updated Table 1. The row labels in the updated table state that the errors represent the global mean.

*Section 3, paragraph 3 and related figures (4 and 5): Wouldn't the model-produced all-sky flux responses between 2006 and 2096 include cloud feedbacks as well? If so, what is the value of comparing those responses with corresponding all-sky kernel estimates that do not include the cloud changes (as stated on L11)?*

Thank you for prompting us to elaborate on the focus of the validation. As mentioned above, we have overhauled the validation in the revised version of the manuscript. Both clear-sky and all-sky fluxes are calculated by the model. All-sky fluxes do ultimately determine climate feedbacks, but the method that is typically used to calculate the cloud feedback, the adjusted cloud radiative effect technique, results in errors of all-sky and clear-sky fluxes (in $\mathrm{Wm}^{-2}$) which are exactly equal.

*Section 4, L22-23: change "based on changes in TOA..." to "based on model-computed changes in TOA. . ."*

This paragraph no longer appears in the revised manuscript. Instead, the error due to internal variability is diagnosed explicitly in the previous section.

*Section 4, L24: Insert "approximately" or symbols, e.g., "∼", in front of these values.*

As with the previous comment, this paragraph no longer appears in the revised manuscript.

*Section 5, first paragraph: It would be useful to 1) give some brief qualitative discussion about how these calculations are conducted (e.g., that the kernels are first monthly averaged then multiplied by monthly-resolved climatological changes in the fields) and 2) Include here a reference to section 6 where one can find more information about relevant example code.*

Thank you for pointing out how we can make better guide our readers. We have made both of these changes.

*Section 5, L6: Could you clarify why one needs long-term mean water vapor mixing ratios?*

The moisture kernels are normalized by climatological moisture so that the difference follows the logarithm of moisture. The logarithm of moisture changes more linearly with warming than moisture itself, as noted by Soden et al (2008), so this increases the accuracy of the fluxes estimated with kernels over using the unnormalized change in mixing ratio.

*Table 2 caption: Suggest changing the "Here" heading to "CAM5"*

We removed the word "here" (CAM5 already appears earlier in the sentence).

*Section 5, L25: Unsure why long timescales are more similar to your calculations since your kernels are only computed for one year. Please explain.*

Block and Mauritsen (2013)'s "long" timescale is simply the long-term climate response. Their alternative short timescale focuses on the first 20 years of a rapid transient response. The climate change from 2006 to 2096 is expected to be more similar to the developed long-term response from their "long" timescale, and thus it is the appropriate comparison here.

The kernels are only calculated for one year (in our kernels as well as those from Block and Mauritsen 2013), rather than, say, 30, or even individually for each experiment, because the computational expense and the human expense of managing the calculation is large. The purpose of radiative kernels is to provide a computationally efficient short-cut to decompose radiative fluxes into their contributions from temperature, moisture, and albedo.

We now clarify in the text that the long timescale refers to the application of the kernels, rather than the kernels themselves.

*Section 5, L30, with regard to "0.02," perhaps also specify which feedback this is and give this value as a percentage of the mean as well. I also recommend reiterating here that this estimate does not account for potential variability among kernels themselves if computed from different years/ensemble members.*

This paragraph no longer appears in the revised manuscript. Instead, we diagnose the error due to internal variability explicitly using the ensemble in the revised Section 3.

*Section 6, L4: Please again specify what the four radiative kernels are. Also note that they are provided as monthly averages in the files.*

We now include this and also state what the two forcings are.

*Typos/grammar:*

Thank you for your careful reading of the manuscript. We have made all of the recommended changes.

*Section 2, page 2, L25: change "has units" to "have units"*

*Section 3, page 4, L5: insert "of" between "decomposition" and "the"*

*Section 3, page 4, L6: Suggest changing "linear in. . ." to "linear with respect to. . ."*

*Section 5, page 5, L9-10: Should "the changes in top-of-atmosphere radiative feedbacks" just be "the top-of-atmosphere radiative feedbacks"?*

*Section 5, page 5, L16: "m(Pendergrass, 2017a)odel"*

[Figure]

**2 Response to Anonymous Reviewer 2**

We thank Reviewer 2 for their thorough and critical feedback on our manuscript. We have incorporated your feedback and expanded the validation, and hopefully improved the manuscript. In this document, your comments are italicized and our responses are in plain type.

*This paper presents a new set of radiation kernels that can be used for feedback analysis. This affords a useful means for GCM inter-comparison and for understanding their differences in climate sensitivity. I recommend publication after the following issues are properly addressed.*

*Two major suggestions are: 1. Provide comparisons between this new set of kernels and other kernels and note their differences. This would serve the community greatly to understand whether (and how) feedback determined using the kernel method is sensitive to the kernel dataset used.*

Comparison between datasets is certainly important and useful. We touch on this briefly in Table 2. Comparison of spatial and vertical structure across different kernels is explored in Shell et al (2008). Furthermore, this is the topic of a separate, forthcoming manuscript, which includes extensive comparison across many kernel datasets, which the first author is involved in. We intend the present manuscript to focus on documenting the CAM5 kernels, consistent with the scope of ESSD, leaving comparison among kernels to the forthcoming manuscript. That said, in the process of preparing that manuscript, we have made comparisons with other existing kernels and are confident that our kernels do compare reasonably with others, despite that this analysis is not included in the present manuscript. Also of potential relevance to your comment, see below for how the zonal mean kernels as shown in Figs. 1 and 2 now are much more consistent with previously published kernels, after resolving a plotting error.

*2. Document more extensive validation tests. Currently only global mean values from one case (2096-2006) are reported (Fig. 4, 5); it is not clear whether the percentage errors reported are representative. Error statistics based on global maps of radiation changes and time series of global means would make a more rigorous assessment.*

In the revised version of the manuscript, we have overhauled and extended the validation. Because the radiative forcing varies over time in the RCP8.5 scenario that was used to generate the kernels, validating the kernels themselves (rather than the forcing fields, which are secondary) is only tractable for the years 2006 and 2096. But we now make use of the other 39 members of the CESM1 large ensemble to validate against the changes from 2006 to 2096 in the members that were not used to generate the kernels. In Fig. 3 of the revised manuscript, we document the global-mean absolute error of TOA and surface LW and SW changes in radiative flux for each of the ensemble members. We have updated Table 1 to document the error of global mean kernel-estimated flux change and global mean absolute error of kernel-estimated flux changes from ensemble member 1.

*Additional comments*

*Page 1, Line 21. It is not a correct statement. New TOA and surface kernels computed from EARi[sic] atmosphere have been made available by Huang et al. (2017). This should be referenced here.*

Thank you for pointing out this omission. In the revision, we qualify that Previdi (2010) are the only model-based surface kernels, but Huang et al (2017) provide reanalysis-based kernels.

*Page 2, Line 7. Note there is strong inter-annual variations of atmospheric states, e.g., El Nino vs. La Nina, which affects kernel values quantitatively. In relation to Suggestion 2 above, it is worth discussing, and if possible demonstrating, (in)accuracy in feedback, e.g., in the central tropical Pacific region, related to this issue.*
In Fig. 4 of the revised manuscript, we show maps of the mean error across these 39 ensemble members to document the spatial pattern of error.

*Fig. 1. I am surprised to see the temperature sensitivity maximize in the middle tropical troposphere (âĹij600hPa) here, which is noticeably different from other kernel datasets (e.g., Soden et al. 2008, Huang et al. 2017) - that may be due to misplaced clouds in CAM?*

Please see response below.

*Fig. 2. I am also surprised by that the surface radiation sensitivity to temperature and humidity doesn't maximize in the lowermost atmospheric layer. These results do not agree with radiative transfer-based expectations.*

Thank you for pointing out these inconsistencies with existing literature. Upon revisiting our code (and at the prompting of kernel user Paulo Ceppi), we realized that the way we incorporated surface pressure into the grid before taking the zonal average was the cause of this difference. We have updated Figures 1 and 2 accordingly, and they are now much more in line with other published zonal-mean kernels. The bug had a small effect in one script in the provided software, which we have now updated as well. To be clear, this was only a plotting error and has no effect on the kernels themselves, nor on any results in hybrid-sigma coordinates (Figs. 3-6 and Tables 1 and 2).

Please also note the supplement to this comment: https://www.earth-syst-sci-data-discuss.net/essd-2017-108/essd-2017-108-AC1-supplement.pdf

[Figure]

**Supplement:**

**Surface and top-of-atmosphere radiative feedback kernels for CESM-CAM5**

Angeline G. Pendergrass[1], Andrew Conley[1], Francis Vitt[1]

[1]National Center for Atmospheric Research, Boulder, CO, 80305, USA

5    *Correspondence to*: Angeline G. Pendergrass (apgrass@ucar.edu)

[revised manuscript text omitted]

**2.1 Radiative kernels**

Together, all calculations consumed approximately 200,000 core hours on NCAR's Yellowstone supercomputer. The limiting factor for throughput is the size of PORT input data: three-hourly 3-D temperature, moisture, and cloud fields; about 7.5 TB of disk space are needed to run one month of three-hourly PORT calculations for each vertical level of each kernel, with 63 global radiative transfer computations per kernel-month (control, surface temperature, and albedo each require one kernel-month, and atmospheric temperature and moisture each require 30). The kernel calculation is run for one year. The TOA radiative kernels are shown in Fig. 1, and surface kernels in Fig. 2. The atmospheric column kernel is the difference between these two kernels. The procedure to produce each kernel follows below. To create these plots, the kernels are regridded to standard CMIP5 pressure levels in the troposphere, including pressure weighting (a vertical regridding script is available at https://github.com/apendergrass/cam5-kernels), and then the zonal and annual means are calculated. A description of the physical drivers of the changes in radiative fluxes is available from Ingram (2010) for the TOA and Pendergrass and Hartmann (2014) for the atmospheric column.

[revised manuscript text omitted]

First, we quantify the error of the kernel-estimated change in radiative flux of the ensemble member from which the kernels are computed. The changes in radiative flux associated with the temperature, water vapor, and albedo feedbacks are calculated using the changes in monthly-mean model fields, and then the change in radiative forcing is added. The global mean modelled and kernel-estimated changes in radiative flux as well as the error in global mean and global-mean absolute error are documented in Table 1. For the global-mean absolute error, the change in annual mean radiative fluxes are calculated, then the absolute value of the error for each ensemble member is calculated at each grid point, and finally the global mean of this quantity is calculated. These error estimates include errors due to sampling every 3 hours (instead of at each model timestep) and due to the nonlinearity inherent in the kernel method. While we only document the changes in clear-sky radiative flux response in Table 1, the errors in all-sky radiative fluxes are exactly the same when cloud feedback is estimated using the adjusted cloud radiative effect method described in (Soden et al., 2008). Errors in global-mean radiative flux change range from $0.1 - 0.8$ Wm$^{-2}$, while global-mean absolute errors range from 0.7-1.4 Wm$^{-2}$. Errors in SW fluxes are smaller than for LW fluxes.

Because the kernel calculation is computationally intensive, it is based on just one year. Next, we quantify the error of the kernel-estimated radiative fluxes compared to members 2-40 of the CESM1 large ensemble. This error estimate includes the

effect of our choice of a single year, as well as nonlinearity and 3-hourly sampling. The global-mean error for each ensemble member is shown in Fig. 4. The global-mean absolute errors are not especially larger for the ensemble mean than for member 1. The change in TOA LW fluxes has more interannual variability than TOA SW and surface fluxes. The error of global mean change in radiative flux is similar to member 1 for SW fluxes but larger for LW fluxes (0.9 Wm$^{-2}$ at both the surface and TOA; not shown). The spatial pattern of ensemble-mean error is shown in Fig. 5. The LW errors, particularly for the surface, are large in the tropics. For the TOA, there are also substantial errors at high latitudes of both hemispheres. The largest errors in the SW are associated with sea ice edges, the movement of which is not captured well by the kernel method. There are also regions of large error associated with tropical clouds, and at the TOA, there is a bias over Antarctica.

In applications that make use of experiments from models other than CESM(CAM5), errors will also arise due to differences in radiative transfer codes between models (e.g., DeAngelis et al., 2015; Soden et al., 2008); we do not quantify these errors explicitly here, but we do compare feedback decompositions made with other radiative kernels from the literature in Section 4.

**4 Application**

Next we show a sample application of the kernels to the CESM large-ensemble simulations. In order to apply the kernels, one needs monthly changes in temperature, mixing ratio, and surface albedo, as well as long-term mean water vapor mixing ratios to calculate the logarithm of water vapor change (Soden et al., 2008). To calculate the cloud feedback, change in cloud radiative effect is also required. Then, the monthly-resolved kernels and changes in atmospheric state are convolved to obtain the changes in radiative flux components. Example code for applying the kernels is available at https://github.com/apendergrass/cam5-kernels.

We apply the radiative kernels to the CESM large ensemble integrations to diagnose the  top-of-atmosphere radiative feedbacks . The changes in surface and tropospheric temperature, water vapor mixing ratio, surface albedo, and cloud radiative effect are calculated from 30-year averages for each month from each ensemble member, 1976-2005 and 2071-2100. The cloud feedback calculation follows Soden et al., (2008). The global, annual mean change in radiative flux for each feedback is calculated and then normalized by the change in global-mean surface temperature for each ensemble member; then, the ensemble average is calculated. Results are shown in Table 2 and Fig. 6.

Table 2 shows a comparison between feedback values calculated here and those diagnosed with other kernels and model simulations. Soden et al., (2008) report radaitive feedbacks from 14 CMIP3 model simulations using three different sets of radiative kernels from CAM3, GFDL, and CAWCR kernels. The Planck response agrees closely. Water vapor and albedo feedbacks are both within 0.2 Wm$^{-2}$K$^{-1}$. The cloud feedback differs by only 0.1 Wm$^{-2}$K$^{-1}$, despite the

fact that Soden et al., (2008) do not account  for aerosol radiative forcing. The only notable disagreement is in the lapse rate feedback by 0.4 Wm$^{-2}$K$^{-1}$. Because the Planck feedback agrees closely between the two calculations, the difference is probably not due to the temperature kernel. Instead, it may be caused by differing upper tropospheric temperature amplification between the CMIP3 and CESM large ensemble simulations or due to underlying differences in the radiation

5 codes.  Block and Mauritsen (2013) report radiative feedbacks using MPI-ESM-LR kernels applied to abrupt carbon dioxide quadrupling experiments with the same model (they compare kernels calculated from different base states and  apply them to short transient response and more developed long timescale response; we compare with their control base state kernels applied to long timescale climate response because this is most similar to our application). There is remarkably close agreement for all feedbacks, excepting only the water vapor feedback. This

10 could arise from the kernels or from the change in water vapor in the simulations.

15

**5 Data and code**

The provided dataset includes the four monthly-mean radiative kernels, atmospheric temperature, surface temperature, water vapor, and  surface albedo, and radiative forcing from greenhouse gases and aerosols. Data are provided on the

20 CESM hybrid-sigma grid for comparison with CESM simulations. The datasets include net all-sky and clear-sky radiative fluxes at both the top-of-atmosphere and surface. The sign convention is the same as CESM's: shortwave fluxes are positive downward, and longwave fluxes are positive upward. Sample temperature, moisture, surface radiative fluxes, and surface pressure are also included, as well as sample code to facilitate use of the kernels.

25 The data and code to calculate TOA temperature, water vapor, and albedo feedbacks are available for immediate download at https://zenodo.org/record/997902 (though without directory structure) and through ESGF at https://www.earthsystemgrid.org/dataset/ucar.cgd.ccsm4.cam5-kernels.html (Pendergrass, 2017). The files included are listed in Table 3. Additional software tools – to regrid the kernels to pressure levels (including CMIP standard levels), calculate TOA Planck, lapse rate, and cloud feedbacks – are available at https://github.com/apendergrass/cam5-kernels.

**6 Path forward**

There is room to improve the accuracy of these and other radiative kernels. Future work could explore new sampling strategies to capture both the diurnal cycle and interannual variability (which would be particularly important for regional applications), directly compare different kernels, and quantify the radiative forcing in climate simulations.

5 *Acknowledgements*

William Frey and Ryan Kramer provided valuable feedback validating the kernels, testing and debugging code. Paulo Ceppi provided valuable feedback on the discussion version of the manuscript as well as the associated software. A.G.P. was supported by an NCAR Advanced Studies Postdoctoral Research Fellowship and by the Regional and Global Climate Modeling Program (RGCM) of the U.S. Department of Energy's, Office of Science (BER), Cooperative Agreement 10 DE-FC02-97ER62402. NCAR is sponsored by the National Science Foundation. Computing resources were provided by the Climate Simulation Laboratory at NCAR's Computational and Information Systems Laboratory (CISL), sponsored by the National Science Foundation and other agencies.

**Table 1:** Validation. Top-of-atmosphere (TOA) and surface clear-sky radiative responses (Wm$^{-2}$) for 2096-2006 from CESM1 large ensemble member 1 (from which the kernels are derived) and the estimated radiative fluxes.  using the kernels and radiative forcing.

| | TOA | | Surface | |
|---|---|---|---|---|
| | LW  | SW | LW | SW |
| Member 1 response | -1.3 | 3.1 | 9.5 | -1.4 |
| Kernel estimate | -0. | 3.4 | 10.3 | -1.2 |
| Error of global mean | 0.3 | 0.3 | 0.8 | 0.1 |
| Global mean abs. error | 1.4 | 0.2 | 1.4 | 1.3 |

**Table 2: Comparison of TOA radiative feedbacks.** TOA radiative feedbacks (Wm$^{-2}$K$^{-1}$) averaged over 40 CESM large ensemble simulations diagnosed with CAM5 radiative kernels , compared against those from CMIP3 model simulations diagnosed with three different kernels as reported by Soden et al., (2008), and MPI-ESM-LR control state kernels and years 21-150 of abrupt carbon dioxide quadrupling simulations from the same model (Block and Mauritsen, 2013).

| Feedback | Here | Soden et al., (2008) | Block and Mauritsen (2013) |
|---|---|---|---|
| Planck | -3.2 | -3.1 or -3.2 | -3.19 |
| Lapse rate | -0.58 | -1 | -0.64 |
| Water vapor | 2.1 | 1.9 | 1.79 |
| Albedo | 0.51 | 0.3 | 0.48 |
| Cloud | 0.66 | 0.77 | 0.62 |

**Table 3: Included data files.** Data files comprising the dataset.

| Filename | Size | Units | Description |
|---|---|---|---|
| alb.kernel.nc | 20 MB | W m$^{-2}$ %$^{-1}$ | Albedo kernel |
| ts.kernel.nc | 20 MB | W m$^{-2}$ K$^{-1}$ | Surface temperature kernel |
| t.kernel.nc | 608 MB | W m$^{-2}$ K$^{-1}$ level$^{-1}$ | Air temperature kernel |
| q.kernel.nc | 1.2 GB | W m$^{-2}$ K$^{-1}$ level$^{-1}$ | Moisture kernel |
| ghg.forcing.nc | 41 MB | W m$^{-2}$ | Greenhouse gas forcing |
| aerosol.forcing.nc | 41 MB | W m$^{-2}$ | Aerosol forcing |

| PS.nc | 5.1 MB | Pa | Surface pressure |

[Figure]

[Figure]

**Figure 1: Top-of-atmosphere kernels from CESM1(CAM5).** Zonal, annual mean temperature, longwave moisture and shortwave moisture kernels for all-sky and clear-sky. In panels (e) and (g) all-sky kernels are shown in solid and clear-sky dashed. The sign convention is positive downward.

[Figure]

[Figure]

**Figure 2: Surface kernels from CESM(CAM5).** Zonal, annual mean temperature, longwave moisture and shortwave moisture kernels for all-sky and clear-sky. In panels (e) and (g) all-sky kernels are shown in solid and clear-sky dashed. The sign convention is positive downward.

**CESM(CAM5) radiative forcing, 2096-2006, W m$^{-2}$**

[Figure]

**Radiative forcing, 2096-2006 (W m$^{-2}$)**

[Figure]

**Figure 3: Radiative forcing.** Net (LW+SW) radiative forcing under the RCP8.5 scenario diagnosed from CESM(CAM5) for greenhouse gases (left) and the direct aerosol radiative forcing (right) at the TOA (top) and surface (bottom).

[Figure]

**Figure 4: Validation across ensemble members.**  of  kernel-estimated clear-sky  radiative  flux change from 2006 to 2096 for members 2-40 of the CESM1 large ensemble for LW (left) and SW (right) fluxes at the TOA (top) and surface (bottom).

[Figure]

Figure 5: Validation of surface kernels. Surface clear-sky and all-sky radiative field validation (excluding cloud contributions).

**Mean error of kernel-estimated response for ens. members 2-40 (W m$^{-2}$)**

[Figure]

**Figure 5: Spatial pattern of error.** Mean error of kernel-estimated radiative flux change from 2006 to 2096 for members 2-40 of the CESM1 large ensemble for LW (left) and SW (right) fluxes at the TOA (top) and surface (bottom).

[Figure]

**Figure 6: CESM Large Ensemble kernels.** Feedback calculation for the CESM 40-member large ensemble using the TOA kernels.